# Modeling of a Microscale Surface Using NURBS Technique

**Jeongki Jang and Kyungmok Kim \*** 

School of Aerospace and Mechanical Engineering, Korea Aerospace University, 76 Hanggongdaehak-ro, Deogyang-gu, Goyang-si, Gyeonggi-do 412-791, Korea; jjk05111@naver.com

\* Correspondence: kkim@kau.ac.kr; Tel.: +82-2-300-0288

**Abstract:** This article describes microscale surface modeling using the Non-Uniform Rational B-Spline (NURBS) surface interpolation technique. A three-dimensional surface model was generated on the basis of measured surface profile data. To validate this model, three brass specimens having different roughness values were used. Direct comparison between measured profiles and the curves modeled with NURBS was employed. It was identified that the proposed method allows the generation of microscale models similar to actual surfaces. Finally, a method to extract the Bearing Area Curve (BAC) from a 3D model was detailed. The proposed modeling will be useful for the characterization of bearing capacity of the surface and for contact analysis.

**Keywords:** roughness; surface topography; NURBS surface interpolation

## 1. Introduction

Surface roughness is an important parameter of friction and wear. In order to evaluate friction and wear, it is necessary to understand the topography of a contact surface. Modeling of surface topography has been developed with several methods, including the statistical approach and the fractal approach.

In the statistical approach, modeling is employed with statistical roughness parameters. The pioneer of the numerical statistical approach is the GW model proposed by Greenwood and Williamson [1]. This model describes all asperities as a hemispherical shape that maintains the same curvature. The statistical approach is being developed on the basis of GW model [2–5]. These approach models contain not only amplitude parameters such as center-line average ($R_a$), root mean square ($R_q$) and standard deviation ($\sigma$), but also spacing parameters such as peak density ($N_p$) and zero crossing density ($N_0$). These methods allow fast and simple modeling of surface topography. However, the biggest disadvantage of the methods is that it depends on sample length.

The fractal approach overcame the disadvantage of the statistical model by using multi-scale to minimize the effect of measurement intervals. Majumdar and Bhushan, and Majumdar and Tien proposed fractal models using self-affinity characteristics of an actual surface [6,7]. This model uses parameters with independent length scales such as fractal dimension. Because of this advantage, there are several studies on fractal model approaches [8,9]. However, the fractal model excludes an interaction by deformation between close asperities under high loading. For micromechanical contact modeling, the resolution dependences of the plasticity index and of the contact prediction by a fractal model were investigated [10].

In order to overcome the limitations of fractal and statistical approaches, methods using measured data were developed [11–13]. Aramaki et al. proposed a method describing asperities using a parabola [11]. Ciulli et al. and Wen et al. developed this method for the purpose of a 2D surface

profile modeling using real data [12,13]. However, little is found on accurate methods to create a three-dimensional surface profile similar to an actual micro-surface profile.

Non-uniform rational basis spline (NURBS) is a precise mathematical model for generating surfaces. In this paper, a three-dimensional surface model is developed using the NURBS surface interpolation technique. Micro-surface profiles of brass specimens having different roughness were measured with a commercial surface profiler. Direct comparison was then conducted between measured surface profiles and those predicted by the proposed model. Finally, a three-dimensional model was generated as vendor-neutral file format (e.g., the Initial Graphics Exchange Specification (IGES) file). Bearing Area curves (BAC) were drawn from the three-dimensional model.

## 2. Theory

In this paper, a micro-surface is described using NURBS surface interpolation. NURBS curve and surface can describe complex shapes by using control points, weights, and the knots vector [14]. The notation is given in Appendix B.

### 2.1. General Forms of a NURBS Curve and a NURBS Surface

A $p$-degree NURBS curve is defined as:

$$\mathbf{C}(u) = \frac{\sum_{i=0}^{m} w_i \mathbf{P}_i N_{i,p}(u)}{\sum_{i=0}^{m} w_i N_{i,p}(u)}, \tag{1}$$

where, $\{P_i\}$ are the $(m+1)$ control points which are $P_0$ to $P_m$, $\{w_i\}$ are the weights, $\{N_{i,p}(u)\}$ are the nonrational $B$-Spline basis function of $p$-degree. In this paper, the range of the parameter $u$ was set as $0 \leq u \leq 1$, and the weight $w_i$ was defined as unity.

Knots vector is defined as:

$$\boldsymbol{U} = \left\{ a, \cdots, a, u_{p+1}, \cdots u_{t-p-1}, b, \cdots, b \right\}. \tag{2}$$

In Equation (2), $p$ is the degree of $B$-Spline. The values from $u_0$ to $u_p$ are pre-described as $a$, and those from $u_{t-p}(= u_{m+1})$ to $u_t$ were $b$. That is, the numbers of $a$ and $b$ are equal to $p+1$. In this paper, because of the range of the parameter $u$ ($0 \leq u \leq 1$), the values of $a$ and $b$ were set to zero and unity, respectively. Using the knots vector, $B$-Spline basis function of $p$-degree is defined as:

$$N_{i,p}(u) = \frac{(u - u_i)N_{i,p-1}(u)}{u_{i+p} - u_i} + \frac{\left( u_{i+p+1} - u \right)N_{i+1,p-1}(u)}{u_{i+p+1} - u_{i+1}}, \tag{3}$$

$$N_{i,0}(u) = f(x) = \begin{cases} 1, & u_i \leq u \leq u_{i+1} \\ 0, & \text{otherwise} \end{cases}, \tag{4}$$

Thus, a NURBS curve can be obtained from Equations (1) and (3).

A NURBS surface is determined according to two directions of $u$ and $v$. It was described as $p$-degree for the $u$-direction and $q$-degree for the $v$-direction. A NURBS surface is defined as:

$$S(u,v) = \frac{\sum_{i=0}^{m} \sum_{j=0}^{n} w_{i,j} P_{i,j} N_{i,p}(u) N_{j,q}(v)}{\sum_{i=0}^{m} \sum_{j=0}^{n} w_{i,j} N_{i,p}(u) N_{j,q}(v)}, \tag{5}$$

where, $\{P_{i,j}\}$ are $(m+1) \times (n+1)$ control points net which is $P_{0,0}$ to $P_{m,n}$, $\{w_{i,j}\}$ are weights, $\{N_{i,p}(u)\}$ and $\{N_{j,q}(v)\}$ are nonrational $B$-Spline basis functions of $p$-and $q$-degrees, respectively. In this paper, the degrees $p$ and $q$ were set to five. In order to describe a NURBS surface, two directions of knots vector, control points net, and weights need to be determined.

In this paper, knots vectors $U$ and $V$ for a NURBS surface were defined as:

$$U = \left\{0, \cdots 0, u_{p+1}, \cdots u_m, 1, 1 \cdots, 1\right\},\tag{6}$$

$$V = \left\{0, \cdots 0, v_{q+1}, \cdots v_n, 1, 1 \cdots, 1\right\}.\tag{7}$$

*2.2. NURBS Curve Interpolation and Surface Interpolation*

For NURBS curve interpolation, it is necessary to define control points and knots vectors. In $p$-degree NURBS curve interpolation, $(m+1)$ control points $\{P_i\}$ were calculated by using $(m+1)$ measured data points $\{Q_k\}$. The relation between $\{Q_k\}$ and $\{P_i\}$ is given as:

$$Q_k = C(\bar{u}_k) = \sum_{i=0}^{m} P_i R_{i,p}(\bar{u}_i) \quad \text{for } k = 0, \cdots, m.\tag{8}$$

Equation (8) can be changed as the following matrix form:

$$\begin{bmatrix} R_{0,p}(\bar{u}_0) & \cdots & R_{m,p}(\bar{u}_0) \\ R_{0,p}(\bar{u}_1) & & R_{m,p}(\bar{u}_1) \\ \vdots & \ddots & \vdots \\ R_{0,p}(\bar{u}_m) & \cdots & R_{m,p}(\bar{u}_m) \end{bmatrix} \begin{bmatrix} P_0 \\ P_1 \\ \vdots \\ P_m \end{bmatrix} = \begin{bmatrix} Q_0 \\ Q_1 \\ \vdots \\ Q_m \end{bmatrix},\tag{9}$$

where, $\{Q_k\}$ are measured data points from zero to $m$, $\{P_i\}$ are control points to be found. $\{R_{i,p}(u)\}$ are the rational basis functions of $p$-degree, and the function is defined as

$$R_{i,p}(\bar{u}_i) = \frac{N_{i,p}(\bar{u}_i)w_i}{\sum_{j=0}^{n} N_{j,p}(\bar{u}_i)w_j}.\tag{10}$$

In order to obtain the rational basis function $\{R_{i,p}(\bar{u})\}$, the parameter $\bar{u}_i$ must be determined. In this paper, the chord length method was selected to determine the parameter $\bar{u}_i$ [15].

$$\bar{u}_i = \bar{u}_{i-1} + \frac{|Q_i - Q_{i-1}|}{\sum_{j=0}^{m}|Q_j - Q_{j-1}|}, \quad \bar{u}_0 = 0 \text{ and } \bar{u}_m = 1.\tag{11}$$

To create a NURBS curve, knots vector must be determined. The knots vector is defined as follows:

$$U = \left\{0, \cdots 0, u_{p+1}, \cdots u_m, 1, 1 \cdots, 1\right\}, \text{ where, } u_{j+p} = \frac{1}{p}\sum_{i=j}^{j+p-1} \bar{u}_i \text{ for } j = 1, \ldots, m-p.\tag{12}$$

By using Equation (11) and weights $\{w_i\}$, which are defined as unity in this paper, the rational basis functions $\{R_{i,p}(\bar{u})\}$ are determined in Equation (10). With the rational basis functions and $\{Q_k\}$, control points $\{P_i\}$ can be given in Equation (9). Finally, a NURBS curve is described with the determined control points $\{P_i\}$, and knots vector $U$ by using Equation (1).

For NURBS surface interpolation, it is necessary to calculate control points net and two knots vectors including $u$-direction and $v$-direction.

In NURBS surface interpolation of $p$-degree for $u$-direction and of $q$-degree for the $v$-direction, $(m+1) \times (n+1)$ control points net $\{P_{i,j}\}$ were calculated by using $(m+1) \times (n+1)$ measured data points $\{Q_{k,l}\}$. The relation between $\{Q_{k,l}\}$ and $\{P_{i,j}\}$ is given as:

$$Q_{k,l} = S(\bar{u}_k, \bar{v}_l) = \sum_{i=0}^{m}\sum_{j=0}^{n} R_{i,p}(\bar{u}_k)R_{j,q}(\bar{v}_l)P_{i,j} \text{ for } k = 0, \cdots, m \text{ and } l = 0, \cdots, n.\tag{13}$$

Equation (13) can be changed to the following matrix form.

$$
\begin{bmatrix}
Q_{0,0} & \cdots & Q_{0,n} \\
\vdots & \ddots & \vdots \\
Q_{m,0} & \cdots & Q_{m,n}
\end{bmatrix}
=
\begin{bmatrix}
R_{0,p}(\overline{u}_0) & \cdots & R_{m,p}(\overline{u}_0) \\
\vdots & \ddots & \vdots \\
R_{0,p}(\overline{u}_m) & \cdots & R_{m,p}(\overline{u}_m)
\end{bmatrix}
\begin{bmatrix}
P_{0,0} & \cdots & P_{0,n} \\
\vdots & \ddots & \vdots \\
P_{m,0} & \cdots & P_{m,n}
\end{bmatrix}
\begin{bmatrix}
R_{0,q}(\overline{v}_0) & \cdots & R_{0,q}(\overline{v}_n) \\
\vdots & \ddots & \vdots \\
R_{n,q}(\overline{v}_0) & \cdots & R_{n,q}(\overline{v}_n)
\end{bmatrix}.
\tag{14}
$$

Determination of the two parameters ($\overline{u}_k^l$ and $\overline{v}_l^k$) is as follows; First, fix $v$ direction and calculate parameter values $\overline{u}_k^l$ for each $l$ with Equation (11). And then, fix $u$ direction and calculate parameter values $\overline{v}_l^k$ for each $k$ with Equation (11). Parameters $\overline{U}$ and $\overline{V}$ are determined by averaging all $\overline{U}^l$ for $l = 0, \cdots, n$ and $\overline{V}^k$ for $k = 0, \cdots, m$.

$$
\overline{u}_k = \frac{1}{n} \sum_{l=0}^{n} \overline{u}_k^l \quad for\ k = 0, \cdots, m.
\tag{15}
$$

$$
\overline{v}_l = \frac{1}{m} \sum_{k=0}^{m} \overline{u}_l^k \quad for\ l = 0, \cdots, n.
\tag{16}
$$

The knots vectors $\boldsymbol{U}$ and $\boldsymbol{V}$ are determined with Equation (12). By using Equation (13) and measured data points $\{Q_{k,l}\}$, control points net $\{P_{i,j}\}$ are calculated. Finally, a NURBS surface can be obtained from Equation (5).

## 2.3. A 3D Surface Model Using NURBS Surface Interpolation

Figure 1 shows the flow chart of NURBS surface interpolation; in the fixed $u$-direction, the parameter $\overline{u}_k^l$ for each curve is computed with Equation (11). In the same way, in the fixed $v$-direction, the parameter $\overline{v}_l^k$ for each curve is calculated with Equation (11). Then, parameters $\overline{U}$ and $\overline{V}$ are determined by using Equation (15). With the parameters $\overline{u}_k$ and $\overline{v}_l$, knots vector $U$ and $V$ can be determined in Equation (12). For determining $R_{i,p}(\overline{u}_k)$ and $R_{j,q}(\overline{v}_l)$ of Equation (13), $N_{i,p}(\overline{u}_k)$ and $N_{j,q}(\overline{v}_l)$ are calculated by using Equations (3) and (4). Control points net $\{P_{i,j}\}$ by using inverse matrix is calculated in Equation (14). By considering the knots vector $\boldsymbol{U}$ and $\boldsymbol{V}$ in Equations (3) and (4), $N_{i,p}(u_k)$ and $N_{j,q}(v_l)$ are re-given. With control points net $\{P_{i,j}\}$ and nonrational $B$-Spline basis functions of $p$-and $q$-degree, the surface $S(u, v)$ is determined in Equation (5). Finally, a three-dimensional model including the surface $S(u, v)$ is obtained in the format of the Initial Graphics Exchange Specification (IGES) file. The NURBS surface interpolation process was employed via a computational code written in MATLAB R2014a.

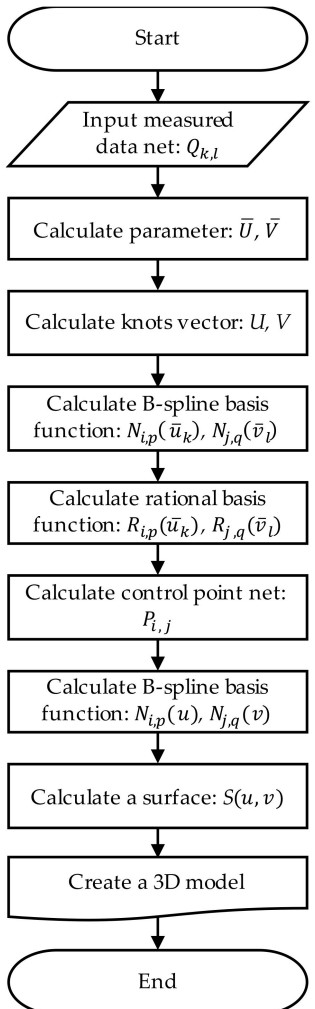

**Figure 1.** Flow chart of Non-Uniform Rational B-Spline (NURBS) surface interpolation.

### 3. Modeling Results and Discussion

To validate this model, three brass specimens were used with different roughness values. Table 1 shows the measured surface roughness. The roughness of surfaces was controlled with an abrasive paper. Surface roughness was measured with a commercial surface profiler (SJ-210, Mitutoyo, Japan).

**Table 1.** Mean and standard deviation ($\sigma$) of surface roughness of tested specimens.

| Specimen Number | | No. 1 | No. 2 | No. 3 |
|---|---|---|---|---|
| $R_a(\mu m)$ | Mean | 1.4980 | 0.5770 | 0.3153 |
| | Standard deviation | 0.1578 | 0.0611 | 0.0506 |
| $R_q(\mu m)$ | Mean | 1.8961 | 0.7374 | 0.3937 |
| | Standard deviation | 0.2114 | 0.0799 | 0.0789 |
| $R_z(\mu m)$ | Mean | 10.3977 | 4.5556 | 2.0727 |
| | Standard deviation | 1.4259 | 0.6346 | 0.5572 |

Table 1 shows the arithmetical average roughness ($R_a$), root mean square ($R_q$), and maximum height ($R_z$) of the specimens. The surface of the first specimen has a high $R_a$ value, while that of the third specimen maintains a low $R_a$ value.

NURBS curve interpolation was applied to the measured 2D profile data. In this modeling, the weight was assumed to be unity and a five-degree B-Spline was used. Figure 2 shows the direct

comparison between measured profiles and modeled curves. An excellent agreement is found between the NURBS curves and the measured 2D profile data.

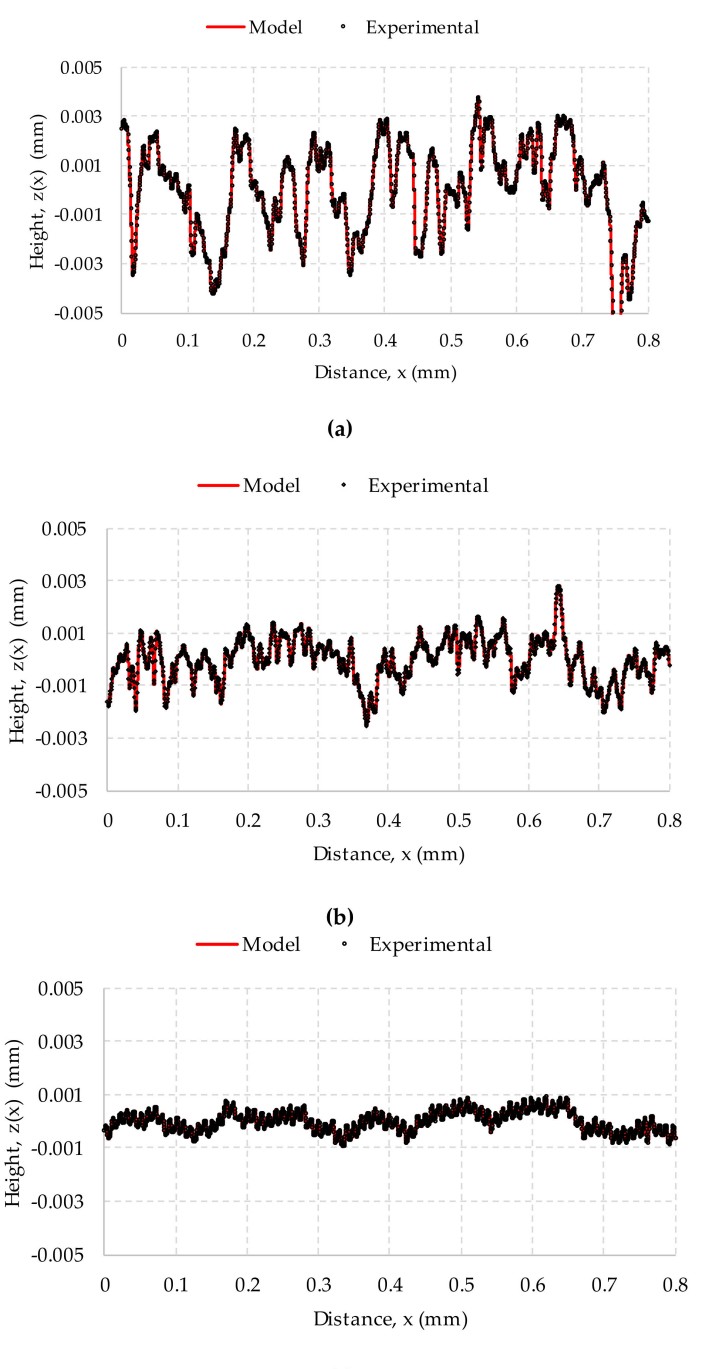

**Figure 2.** Direct comparison between measured and simulated 2D profiles: (**a**) rough specimen (No. 1); (**b**) medium rough specimen (No. 2); (**c**) Smooth specimen (No. 3).

Surface profiles of the three specimens were measured at the interval of 0.01 mm (*y*-axis direction) as shown in Figure 3a, Figure 4a, and Figure 5a. NURBS surface interpolation was then employed with the measured data. Figure 3b, Figure 4b, and Figure 5b show the surfaces modeled with NURBS surface interpolation. It is demonstrated that both rough and smooth surfaces can be generated with the proposed method.

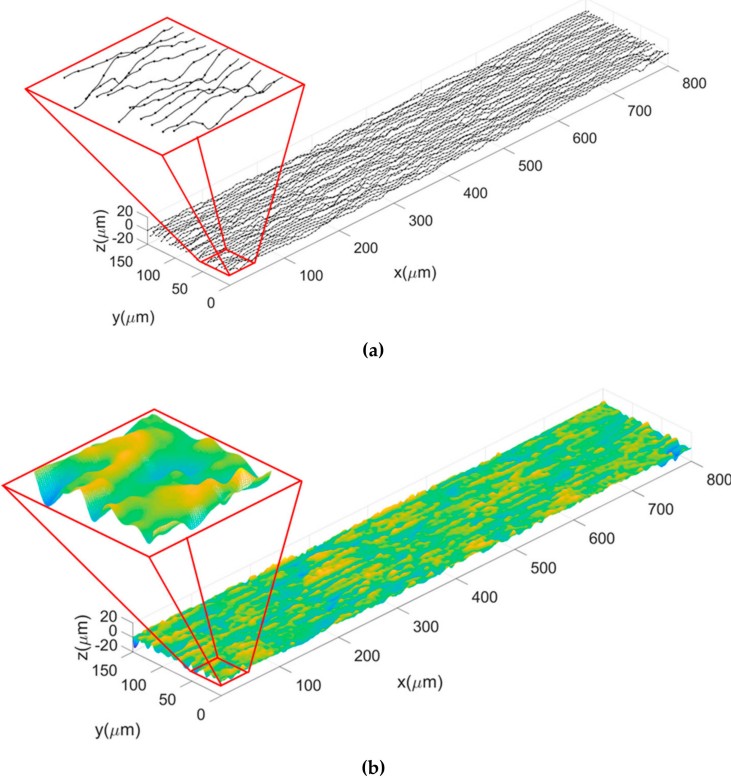

**Figure 3.** Measured profiles and the simulated surface of a rough specimen (Specimen No. 1): (**a**) measured profile data and (**b**) a generated NURBS surface.

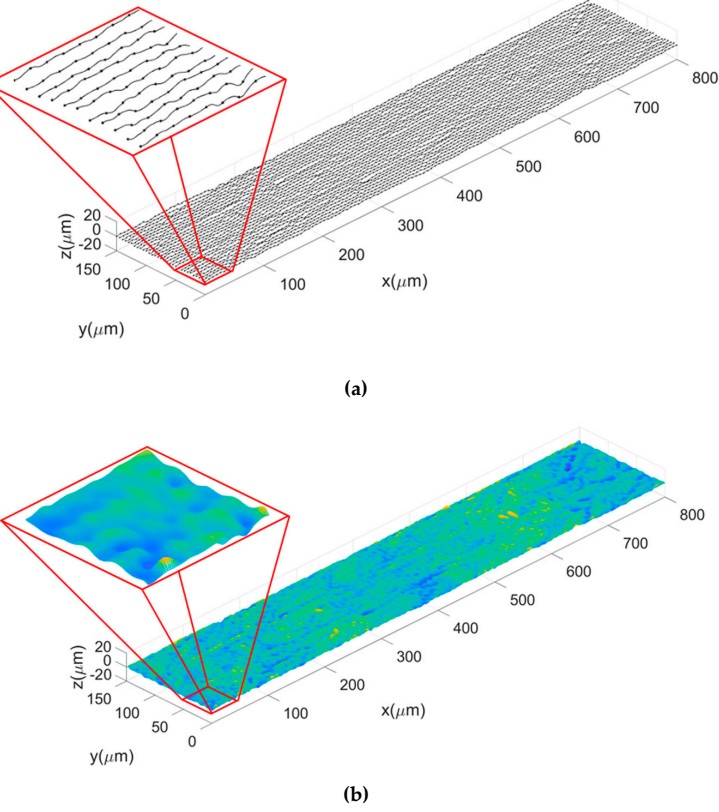

**Figure 4.** Measured profiles and the simulated surface of a medium rough specimen (Specimen No. 2): (**a**) measured profile data and (**b**) a generated NURBS surface.

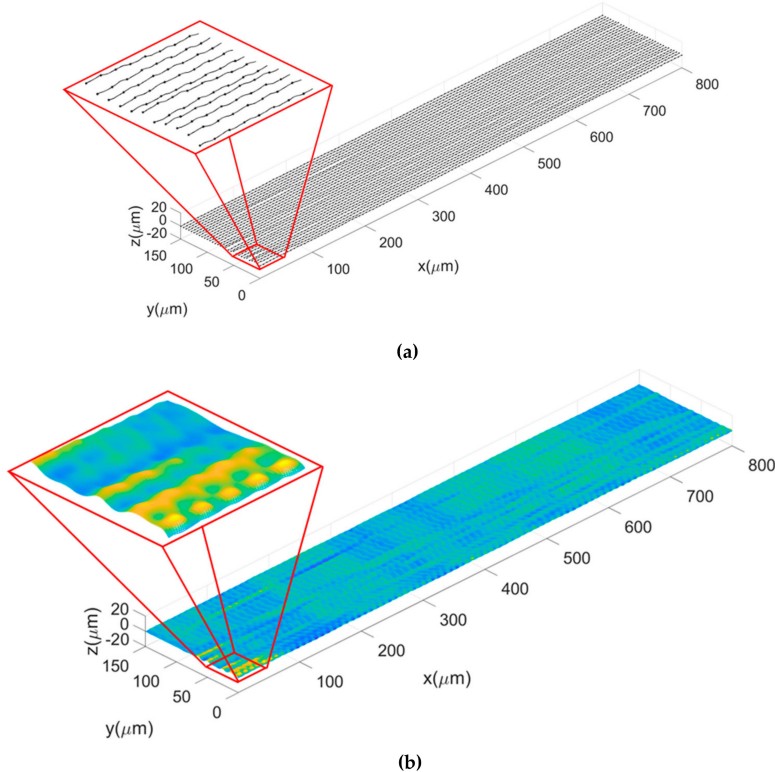

**(a)**

**(b)**

**Figure 5.** Measured profiles and the simulated surface of a smooth specimen (Specimen No. 3): (**a**) measured profile data and (**b**) a generated NURBS surface.

In order to use this model for actual application, a three-dimensional model presented in Figure 6 would be useful. The surfaces modeled with NURBS were transformed as the Initial Graphics Exchange Specification (IGES) file format. The IGES file format allows using the models directly in commercial finite element software such as ABAQUS 2019, ANSYS 2019 and LS-DYNA R8.0.

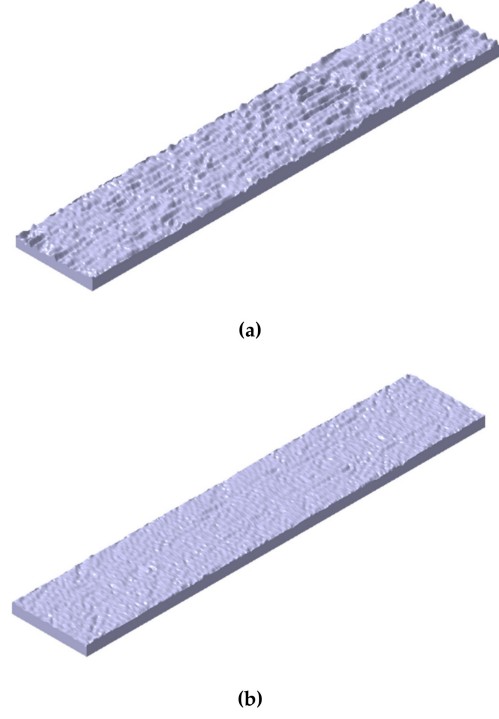

**(a)**

**(b)**

**Figure 6.** *Cont.*

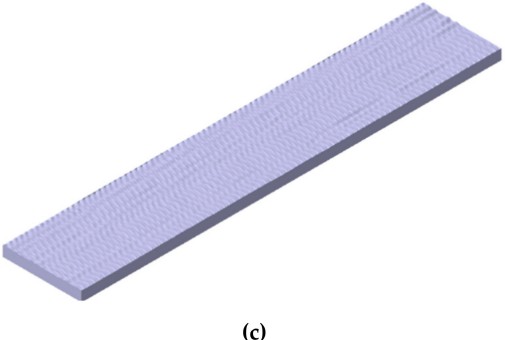

(c)

**Figure 6.** Three-dimensional models generated by the proposed method: (**a**) Rough; (**b**) medium rough; (**c**) smooth surface.

Isogeometric analysis (IGA) is known as the state-of-the-art approach on bridging the gap between Computer-Aided Design (CAD) and Computer-Aided Engineering (CAE) via NURBS technique [16]. The NURBS function for CAD drawing construction can be used as shape functions for creating IGA elements. Thus, the proposed method would apply to isogeometric analysis for obtaining precise geometry and simplified mesh generation.

The proposed modeling allows characterizing the bearing capacity of the surface. In the evaluation of a surface, the bearing area curve (BAC) is often used to describe the distribution of material in the length range of a profile. Thus, in this paper, the bearing area curves of each specimen were drawn with the proposed modeling. Figure 7 shows the flow chart of BAC using the proposed model. The procedure for BAC is as follows: First, extract 2D profile curves from the generated 3D model. In this paper, 31 curves were taken into account for each surface. Second, find the maximum and the minimum height of each curve; then, divide the height into the same interval. Third, calculate the length of a cross line to each height line which is the parallel center line. Lastly, calculate material area ratios (MAR) of each height. There exists detailed information about the calculation of bearing area curve (BAC) in Appendix A.

Figure 8 shows the bearing area curves (BAC) for each 3D model. Thirty-one curves were shown in a single BAC graph. The length between the highest projection and the lowest depression was obviously identified from the graph.

The proposed method using NURBS interpolation can provide a model having a micro-surface similar to that found in an actual surface. Because of using actual measured data, this modeling method could overcome disadvantages of statistical and fractal approaches. In surface evaluation and contact analysis, it is of importance to use an accurate surface model [17,18]. Therefore, the proposed method would be useful for contact analysis. Further work will include direct application to micro-contact analysis found in friction and wear.

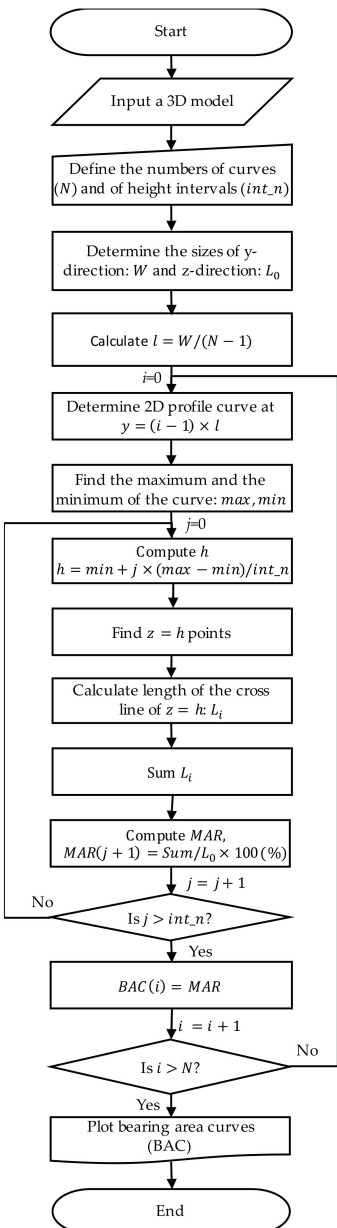

**Figure 7.** Flow chart for determining bearing area curve (BAC). MAR denotes material area ratio.

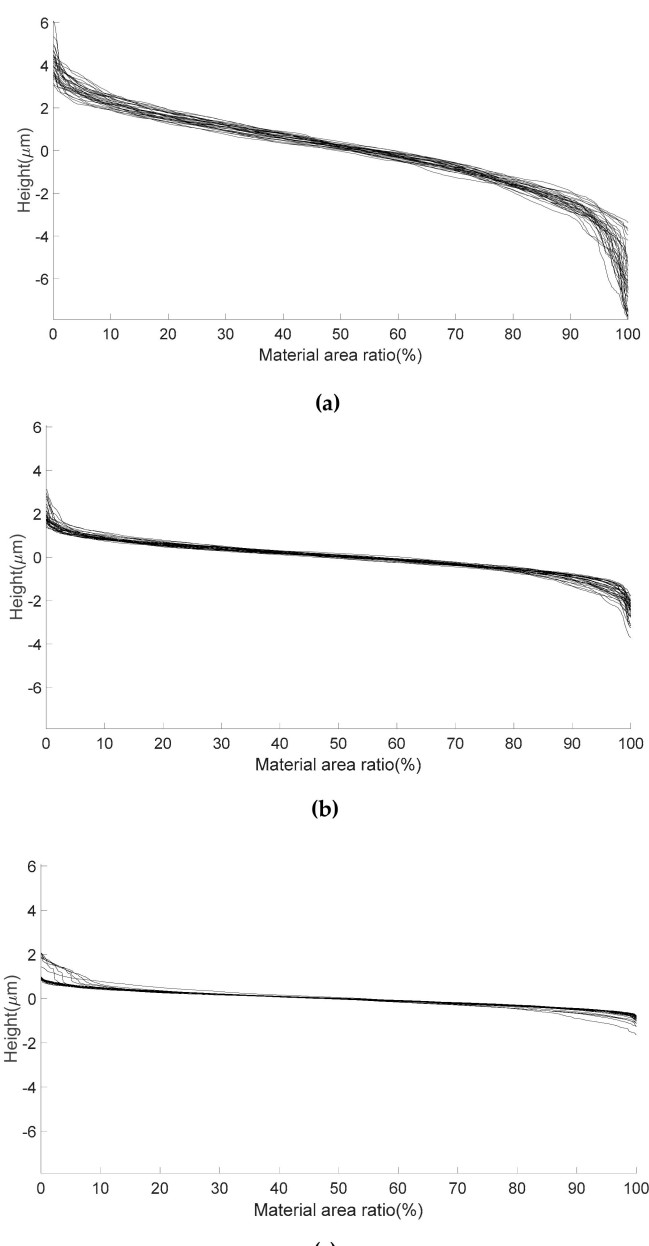

**Figure 8.** The bearing area curves (BAC) determined by the proposed method: (**a**) Rough (Specimen No. 1); (**b**) medium roughness (Specimen No. 2); (**c**) smooth surface (Specimen No. 3).

## 4. Conclusions

In this paper, three-dimensional surface profile modeling was described using NURBS surface interpolation. For the purpose of verification, actual surface profiles of brass specimens were measured. Direct comparison between measured profiles and the surfaces modeled was employed.

The following conclusion was drawn from modeling results.

- The proposed model allows the creation of a curve passing surface roughness points and a single surface including all measured profile data. Thus, the surface similar to an actual surface can be generated on the condition that accurate measurement of the surface profile is carried out. Since the proposed model uses measured profile data, it is possible overcome the limitations of fractal and statistical methods.
- Surface topography was generated with the NURBS surface interpolation method. The surface modeled by the NURBS surface interpolation can be converted as an IGES file format with a 3D

model. This 3D model will be useful for analyzing the contact found in the friction and wear behavior of materials. The method for the characterization of bearing capacity was described from the 3D model; it was identified that Bearing Area Curves (BAC) can be drawn with the proposed model. This method will reduce the time to evaluate the bearing capacity of a surface.

- Future work includes the application of the proposed modeling to contact analysis in sliding, fretting, and lubrication. Particularly, the developed 3D model would be directly used in finite element analysis and an isogeometric analysis of micro-contact problems.

**Author Contributions:** Conceptualization, J.J. and K.K.; software, J.J.; validation, K.K. and J.J.; formal analysis, J.J.; investigation, J.J. and K.K.; writing—original draft preparation, J.J.; writing—review and editing, K.K.; visualization, J.J.; supervision, K.K.; funding acquisition, K.K.

**Funding:** This work was supported by the National Research Foundation of Korea (NRF) grant funded by the Korea government (MSIT) (No. 2019R1A2C1007515)

**Conflicts of Interest:** The authors declare no conflict of interest.

## Appendix A

The bearing area curve (BAC), or Abbott-Firestone Curve, which is set of material area ratios (MAR) is a method of describing surface roughness [19,20]. BAC allows describing the surface roughness graphically by giving information about the peaks and valley's percentage of profile as well as roughness of a core profile. BAC is defined as:

$$MAR(height) = \frac{L_1 + L_2 + \cdots + L_n}{L_0} \times 100 \ [\%], \tag{A1}$$

where, $L_0$ is the total length of a profile, $L_1$ to $L_n$ are the length of the cross line to the height line which is a parallel center line. Figure A1 shows a graphical description to draw the BAC.

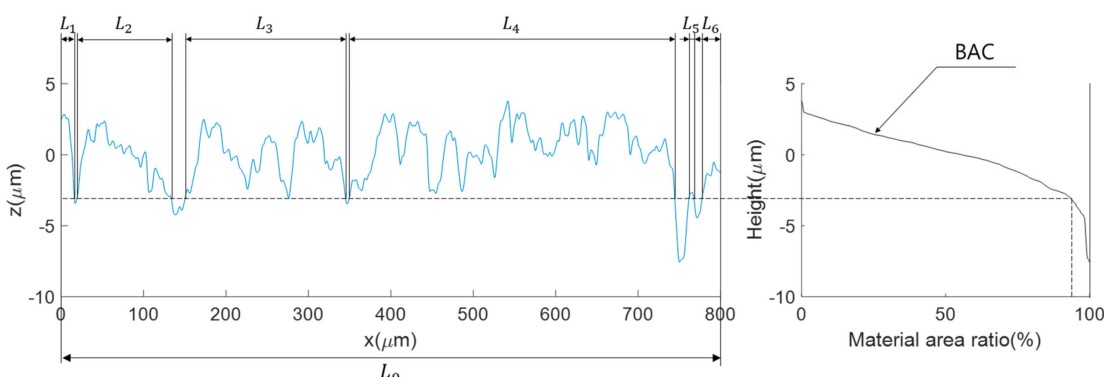

**Figure A1.** Bearing area curve of a profile model.

## Appendix B

| | |
|---|---|
| BAC | bearing area curve |
| $C(u)$ | NURBS curve |
| $N_0$ | zero crossing density |
| $N_{i,p}(u)$ | nonrational B-Spline basis function of $p$-degree |
| $N_p$ | peak density |
| $p, q$ | degree of B-spline |
| $P_i$ | control points |
| $P_{i,j}$ | control points net |
| $Q_i$ | measured profile data |
| $Q_{i,j}$ | measured profile data points net |
| $R_a$ | center-line average roughness |
| $R_{i,p}(u)$ | rational basis functions of $p$-degree |

| | |
|---|---|
| $R_q$ | root mean square roughness |
| $R_z$ | maximum height roughness |
| $S(u, v)$ | NURBS surface |
| $u, v$ | parameter |
| $u_i, v_i$ | element of knots vector |
| $\bar{u}_i, \bar{v}_i$ | approximate parameter value determined by parameterization method |
| $U, V$ | knots vector |
| $w_i$ | weights |
| $w_{i,j}$ | weights net |

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
