# Peer review of "Modeling of a Microscale Surface Using NURBS Technique"

_coatings, doi:10.3390/coatings9120775_

Round 1
Reviewer 1 Report
See attache file

Reviewer 2 Report
1) It needs significant improvement to underlay the necessity of the current work. The main validation is by interpolating an already known set of measured surface data by NURBS. Providing this point does not give a strong added value to the research as it does not compare with any other method, i.e., conventional CAD drawing process.
For example, we can still create an accurate surface function if we have the input of a dense data from the measured surface. Hence, the manuscript requires strong justification on the application od NURBS.
2) There is an important missing information on whether The NURBS interpolation process was carried out via an established software or a computational code developed by the authors. Thus, the authors need to clarify.
3) There is a statement that the IGES file produced can be easily extracted to FEM software, i.e., Abaqus, LS Dyna. This statement is correct; however, there is a lack of discussion on Isogeometric Analysis (IGA).
IGA, currently, is the state-of-the-art on bridging CAD and CAE via spline (NURBS, T spline,etc.). By directly importing the IGES file to FEM software will not improve any current process of FEM post processing. In contrast, via IGA, the NURBS function used for CAD drawing construction can be used as shape functions for creating IGA elements. The authors should emphasise and elaborate further on this matter.
